META-RESEARCH ARTICLE

# A field-wide assessment of differential expression profiling by high-throughput sequencing reveals widespread bias

**Taavi Päll[1], Hannes Luidalepp[2], Tanel Tenson[3], Ülo Maiväli** [3]*

**1** Institute of Biomedicine and Translational Medicine, University of Tartu, Tartu, Estonia, **2** Quretec, Tartu, Estonia, **3** Institute of Technology, University of Tartu, Tartu, Estonia

* ymaivali@gmail.com

## Abstract

We assess inferential quality in the field of differential expression profiling by high-throughput sequencing (HT-seq) based on analysis of datasets submitted from 2008 to 2020 to the NCBI GEO data repository. We take advantage of the parallel differential expression testing over thousands of genes, whereby each experiment leads to a large set of $p$-values, the distribution of which can indicate the validity of assumptions behind the test. From a well-behaved $p$-value set $\pi_0$, the fraction of genes that are not differentially expressed can be estimated. We found that only 25% of experiments resulted in theoretically expected $p$-value histogram shapes, although there is a marked improvement over time. Uniform $p$-value histogram shapes, indicative of <100 actual effects, were extremely few. Furthermore, although many HT-seq workflows assume that most genes are not differentially expressed, 37% of experiments have $\pi_0$-s of less than 0.5, as if most genes changed their expression level. Most HT-seq experiments have very small sample sizes and are expected to be underpowered. Nevertheless, the estimated $\pi_0$-s do not have the expected association with N, suggesting widespread problems of experiments with controlling false discovery rate (FDR). Both the fractions of different $p$-value histogram types and the $\pi_0$ values are strongly associated with the differential expression analysis program used by the original authors. While we could double the proportion of theoretically expected $p$-value distributions by removing low-count features from the analysis, this treatment did not remove the association with the analysis program. Taken together, our results indicate widespread bias in the differential expression profiling field and the unreliability of statistical methods used to analyze HT-seq data.

## Introduction

Over the past decade, a feeling that there is a crisis in experimental science has increasingly permeated the thinking of methodologists, captains of industry, working scientists, and even the lay public [1–6]. This manifests in poor statistical power to find true effects [7,8], in poor reproducibility (defined as getting identical results when reanalyzing the original data by the

**Data Availability Statement:** The authors confirm that all data underlying the findings are fully available without restriction. The code to produce the raw dataset is available on the rstats-tartu/geo-

htseq Github repo (https://github.com/rstats-tartu/geo-htseq). The raw dataset produced by the workflow is deposited in Zenodo https://zenodo.org with doi: 10.5281/zenodo.7529832 (https://doi.org/10.5281/zenodo.7529832). The code to produce the article's figures and models is deposited on the rstats-tartu/geo-htseq-paper Github repo (https://github.com/rstats-tartu/geo-htseq-paper). Individual model objects are deposited in G-Node with doi: 10.12751/g-node.p34qyd (https://doi.org/10.12751/g-node.p34qyd). Code and workflow used to run and analyze RNA-seq simulations are deposited in Zenodo with doi: 10.5281/zenodo.4463804 (https://doi.org/10.5281/zenodo.4463804).

**Funding:** This work was supported by the European Regional Development Fund through the Centre of Excellence in Molecular Cell Engineering (2014-2020.4.01.15-0013 for ÜM and TT) and by grants from the Estonian Research Council (PRG335 for ÜM and TT; PUT1580 for TP). The funders had no role in study design, data collection and analysis, decision to publish, or preparation of the manuscript.

**Competing interests:** The authors have declared that no competing interests exist.

**Abbreviations:** CPM, counts per million; DE, differential expression; FDR, false discovery rate; HT-seq, high-throughput sequencing.

original analytic workflow), and in poor replicability (defined as getting similar results after repeating the entire experiment) of the results [9]. The proposed reasons behind the crisis include sloppy experimentation, selective publishing, perverse incentives, difficult-to-run experimental systems, insufficient sample sizes, overreliance on null hypothesis testing, and much-too-flexible analytic designs combined with hypothesis-free study of massively parallel measurements [10–15]. Although there have been attempts at assessing experimental quality through replication of experiments, prohibitive costs and theoretical shortcomings in analyzing concordance in experimental results have encumbered this approach in biomedicine [16–19]. However, results from a recent replication of 50 experiments from 23 high-profile preclinical cancer studies indicate that ca 90% of effect sizes were overestimated and that over half of the published effects were either in the wrong direction or were wrongly assigned to the non-null category [20].

Another way to assess the large-scale quality of a science is to employ surrogate measures for quality that can be more easily obtained than the full replication of a study. The most often used measure is technical reproducibility, which involves checking for code availability and running the original analysis code on the original data. Although the evidence base for reproducibility is still sketchy, it seems to be well below 50% in several fields of biomedicine [18]. However, as there are many reasons why a successful reproduction might not indicate a good quality of the original study or why an unsuccessful reproduction may not indicate a bad quality of the original study, the reproducibility criterion is insufficient.

Another proxy for quality can be found in published $p$-values, especially the distribution of $p$-values [21]. In a pioneering work, Jager and Leek extracted ca. 5,000 statistically significant $p$-values from abstracts of leading medical journals and pooled them to formally estimate, from the shape of the ensuing $p$-value distribution, the Science-Wide False Discovery Rate or SWFDR as 14% [22]. However, as this estimate rather implausibly presupposes that the original $p$-values were calculated uniformly correctly and that unbiased sets of significant p-values were obtained from the abstracts, they subsequently revised their estimate of SWFDR upwards, as "likely not >50%" [23]. For observational medical studies, by a different method, a plausible estimate for field-wide false discovery rate (FDR) was found to be somewhere between 55% and 85%, depending on the study type [24].

While our work uses published $p$-values as evidence for field-wide quality and presupposes access to unbiased complete sets of unadjusted $p$-values, it does not pool the $p$-values across studies, nor does it assume that they were correctly calculated. In fact, we assume the opposite and do a study-by-study analysis of the quality of calculating $p$-values. This makes the quality of the $p$-value a proxy for the quality of the experiment and the scientific inferences based on these $p$-values. We do not see our estimate of the fraction of poorly calculated $p$-values as a formal quality metric but merely hope that by this measure, we can shed some light on the overall quality of a field.

We concentrate on field of differential expression profiling studies using high-throughput sequencing (HT-seq DE, mostly RNA-seq) for 2 reasons. First, HT-seq has become the gold standard for whole transcriptome gene expression quantification in research and clinical applications [25]. Secondly, due to the massively parallel testing in individual studies of tens of thousands of features per experiment, we can access study-wide lists of $p$-values. From the shapes of histograms of $p$-values, we can identify the experiments where $p$-values were calculated apparently correctly, and from these studies, we can estimate the study-wise relative frequencies of true null hypotheses (the $\pi_0$-s). Also, we believe that the very nature of the HT-seq DE field, where each experiment compares the expression levels of about 20,000 different features (e.g., RNA-s) on average, predicates that the quality of data analysis and specifically statistical inference based on $p$-values must play a decisive part in scientific inference. Simply,

one cannot analyze an HT-seq DE experiment intuitively, without resorting to formal statistical inference. Therefore, quality problems of statistical analysis would very likely directly and substantially impact the quality of science. Thus, we use the quality of statistical analysis as a proxy for the quality of science, with the understanding that this proxy may work better for modern data-intensive fields, where a scientist's intuition has a comparatively more minor role to play.

## Results

### Data mining

We queried the NCBI GEO database for "expression profiling by high-throughput sequencing" (for the exact query string, see Methods), retrieving 43,610 datasets (GEO series) from 2006, when the first HT-seq dataset was submitted to GEO, to December 31, 2020. The yearly new HT-seq submissions increased from 1 in 2006 to 11,604 by 2020, making up 26.6% of all GEO submissions in 2020.

First, we filtered the GEO series containing supplementary processed data files. NCBI GEO database submissions follow MINSEQE guidelines [26]. Processed data are a required part of GEO submissions, defined as the data on which the conclusions in the related manuscript are based. The format of processed data files submitted to GEO is not standardized, but in the case of expression profiling, such files include, but are not limited to, quantitative data for features of interest, e.g., mRNA, in tabular format. Sequence read alignment files and coordinates (SAM, BAM, and BED) are not considered processed data by GEO.

According to our analysis, the 43,610 GEO series contained 84,036 supplementary data files, including RAW.tar archives. After unpacking RAW.tar files, we programmatically attempted to import 647,092 files, resulting in 336,602 (52%) successfully imported files, whereas 252,685 (39%) files were not imported because they were either SAM, BAM, BED, or in other formats most probably not containing $p$-values. We failed to import 57,805 (8.9%) files for various reasons, primarily because of text encoding issues and failure to identify column delimiters.

According to GEO submission requirements, the processed data files may contain raw counts of sequencing reads and/or normalized abundance measurements. Therefore, a valid processed data submission may not contain lists of $p$-values. Nevertheless, we identified $p$-values from 4,616 GEO series, from which we extracted 14,813 unique unadjusted $p$-value sets. While the mean number of $p$-value sets, each set corresponding to a separate experiment, per 4,616 GEO submissions was 3.21 (max 276), 46% of submissions contained a single $p$-value set, and 76% contained 1–3 $p$-value sets. For further analysis, we randomly selected 1 $p$-value set per GEO series.

### P-value histograms

We algorithmically classified the $p$-value histograms into 5 classes (see Methods for details and Fig 1A for representative examples) [27]. The "Uniform" class contains flat $p$-value histograms indicating no detectable true effects. The "Anti-Conservative" class contains otherwise flat histograms with a spike near zero. The "Conservative" class contains histograms with a distinct spike close to one. The "Bimodal" histograms have 2 peaks, one at either end. Finally, the class "Other" contains a panoply of malformed histogram shapes (humps in the middle, gradual increases towards one, spiky histograms, etc.). The "Uniform" and "Anti-Conservative" histograms are the theoretically expected shapes of $p$-value histograms.

We found that overall, 25% of the histograms fall into the anti-conservative class, 9.5% were conservative, 26% bimodal, and 39% fell into the class "other" (Fig 1B). Only 17 of the 4,616

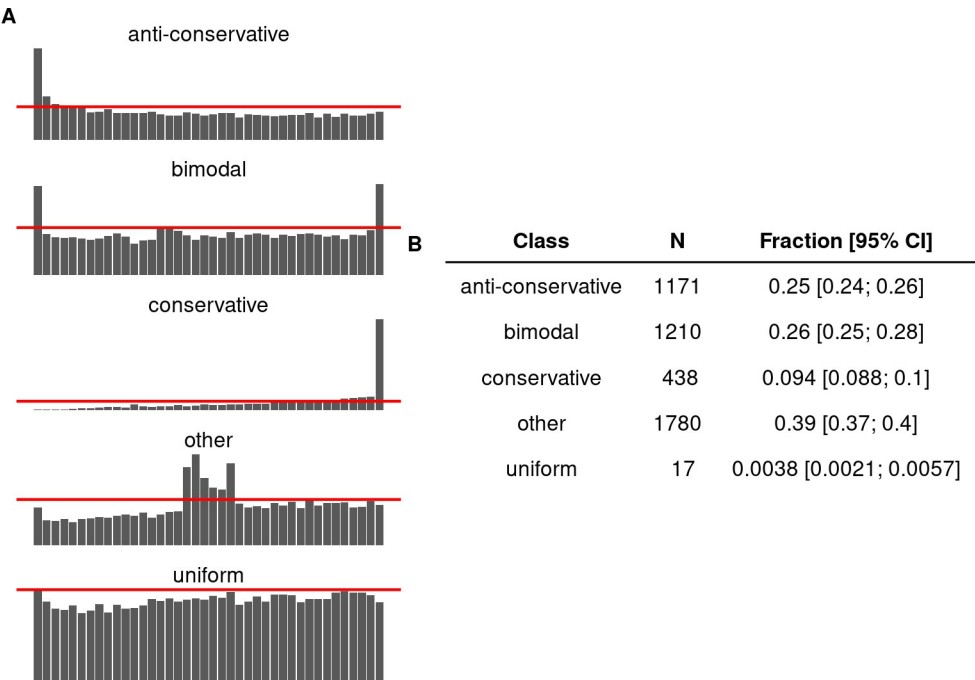

**Fig 1. Classes of *p*-value histograms.** (**A**) Examples of *p*-value histogram classes. Red lines denote the algorithmic threshold separating *p*-value histograms into discrete classes. (**B**) Summary of *p*-value histograms identified from GEO supplementary files. One *p*-value set was randomly sampled from each GEO series where *p*-values were identified. *N* = 4,616; 95% CI are credible intervals. The model object related to panel B can be downloaded from https://gin.g-node.org/tpall/geo-htseq-paper/src/v0.1/models/Class_1.rds.

histograms were classified as "uniform." The median number of features in our sample was 20,954. Interestingly, there are fewer features in anti-conservative *p*-value histograms, as compared to histograms with all other shapes, suggesting different data preprocessing for datasets resulting in anti-conservative histograms (S1 Fig). Logistic regression reveals a clear trend for an increasing proportion of anti-conservative histograms, starting from <10% in 2010 and surpassing 30% in 2020 (S2 Fig). Multinomial hierarchical logistic regression indicates that most differential expression (DE) analysis tools exhibit temporal increases of anti-conservative *p*-value histograms, except for cuffdiff, which has the opposite trend, and glc genomics and deseq, where a clear trend could not be identified (Figs 2A and S3). The increase in the fraction of anti-conservative histograms is accomplished by decreases mainly in the class "other" and "bimodal," depending on the DE analysis tool.

This positive temporal trend in anti-conservative *p*-value histograms suggests improving quality of the HT-seq DE field. Somewhat surprisingly, Fig 2A also indicates that different DE analysis tools are associated with very different proportions of *p*-value histogram classes, suggesting that the quality of *p*-value calculation, and therefore, the quality of scientific inferences based on the *p*-values, depends on the DE analysis tool. We further tested this conjecture in a simplified model, restricting our analysis to 2018 to 2020, the final years in our dataset (Fig 2B). As no single DE analysis tool dominates the field (the top 5 are DESeq2 28%, cuffdiff 27%, edgeR 14%, DESeq 8%, limma 2%; see S4 Fig for temporal trends), a situation where proportions of different *p*-value histogram classes do not substantially differ between analysis tools would indicate lack of tool-generated bias. However, we found that all *p*-value histogram classes, except "uniform," which is mainly unpopulated, depend strongly on the DE analysis tool (Figs 2B, S3, and S5). This effect is quite extreme, extending from nearly zero fraction in

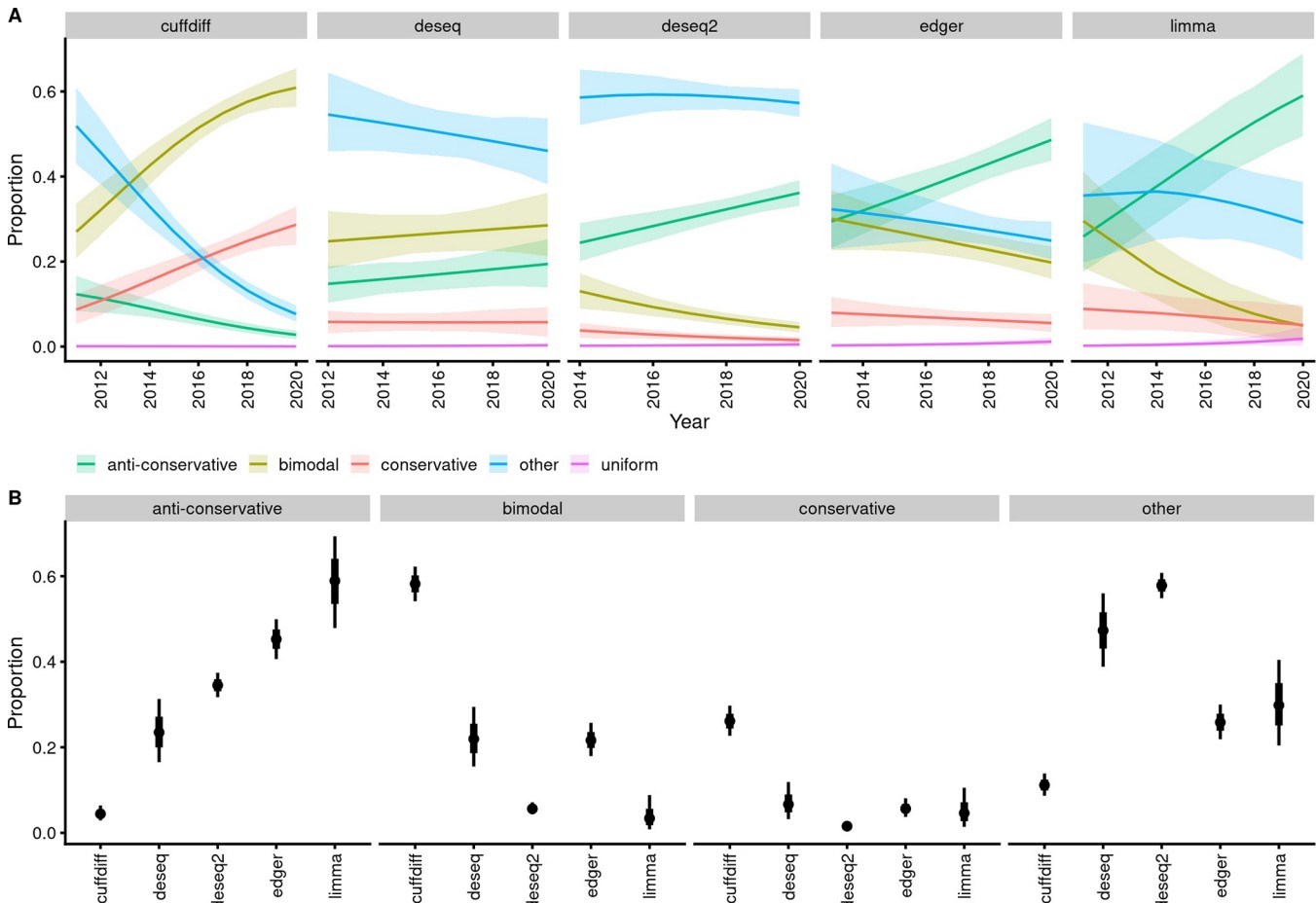

**Fig 2. Association of the *p*-value histogram class with a differential expression analysis tool.** (**A**) Time courses for proportions of different *p*-value histogram classes for the 5 most frequently used DE analysis platforms. Lines denote best fit of the model [class ~ year + (year | de_tool), categorical likelihood]. Shaded areas denote 95% credible regions. $N$ = 4,616. The data file is in S1 Data (**B**) Association of *p*-value histogram type with DE analysis tool; data is restricted to 2018–2020 GEO submissions. Points denote best fit of the model [n | trials(total in de_tool) ~ class + de_tool + class:de_tool, binomial likelihood]. Thick and thin lines denote 66% and 95% credible intervals, respectively. $N$ = 2,930. The model object related to panel A can be downloaded from https://gin.g-node.org/tpall/geo-htseq-paper/src/v0.1/models/Class_year__year_detool_year.rds. The model object related to panel B can be downloaded from https://gin.g-node.org/tpall/geo-htseq-paper/src/v0.2/models/n__trials%28total_in_de_tool%29__Class_de_tool_Class:de_tool_2018up.rds. The data file is in S2 Data. See S3 Fig for a full set of identified DE analysis tools.

cuffdiff to about 0.75 in Sleuth. Using the whole dataset of 14,813 *p*-value histograms—as a check for robustness of results—or adjusting the analysis for GEO publication year, of the taxon (human, mouse, and pooled other), of the RNA source or sequencing platform—as a check for possible confounding—does not change this conclusion (S5B–S5F Fig). The lack of confounding in our results allows a causal interpretation, indicating that DE analysis tools bias the analysis of HT-seq experiments, while the large DE analysis platform-dependent differences suggest a very substantial bias [28].

## The proportion of true null hypotheses

To further enquire into DE analysis tool-driven bias, we estimated from user-submitted *p*-values the fraction of true null effects (the $\pi_0$) for each HT-seq experiment. The $\pi_0$ is a statistic calculated solely from the *p*-values, and it is routinely used as an intermediate quantity needed to fix FDR at the desired level [29]. The quality of FDR control depends on the quality of

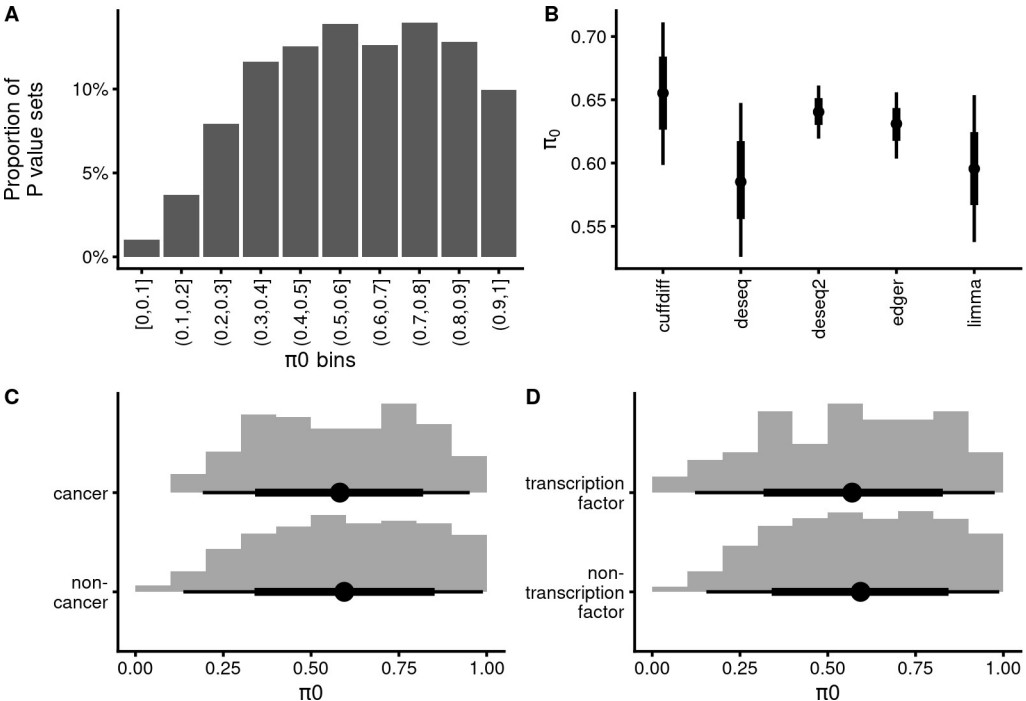

**Fig 3. Association of the proportion of true null effects ($\pi_0$) with DE analysis tool.** (**A**) Histogram of $\pi_0$ values estimated from anti-conservative and uniform *p*-value sets. *N* = 1,188. The data file is in S3 Data. (**B**) Robust linear model [pi0 ~ de_tool, beta likelihood] indicates an association of $\pi_0$ with the DE analysis tool. Points denote best estimates for the mean $\pi_0$ and thick and thin lines denote 66% and 95% credible intervals, respectively. *N* = 1,188. The data file is in S4 Data. (**C**) Histogram of $\pi_0$ values in GEO cancer studies compared to non-cancer studies. The data file is in S5 Data. (**D**) Histogram of $\pi_0$ values in GEO transcription factor studies compared to non-TF studies. The data file is in S6 Data. The model object related to panel B can be downloaded from https://gin.g-node.org/tpall/geo-htseq-paper/src/v0.1/models/pi0_detool_sample.rds.

estimation of the $\pi_0$, which in turn depends on the quality of underlying *p*-values. Thus, the $\pi_0$ can be seen as an estimate of the true proportion of true nulls or as a single number summary of the *p*-value distribution, and in the latter capacity, it can be used as a quality check of the *p*-values used for FDR control of the experiment.

As non-anti-conservative sets of *p*-values (excepting the "uniform") already indicate low quality of both the $\pi_0$ estimate, and of the ensuing FDR control [30], we only calculated the $\pi_0$ for datasets with anti-conservative and uniform *p*-value distributions (*N* = 1,188). Nevertheless, the $\pi_0$-s show an extremely wide distribution, ranging from 0.999 to 0.02. Remarkably, 37% of the $\pi_0$ values are smaller than 0.5, meaning that, according to the calculated *p*-values, in those experiments, over half of the features are estimated to change their expression levels upon experimental treatment (Fig 3A). Conversely, only 23% of $\pi_0$-s exceed 0.8, and 9.9% exceed 0.9. The peak of the $\pi_0$ distribution is not near 1, as might be expected from experimental design considerations, but there is a broad peak between 0.5 and 0.8 (median and mean $\pi_0$-s are both at 0.59). Depending on the DE analysis tool, the mean $\pi_0$-s range over 20 percentage points, from about 0.45 to 0.65 (Fig 3B, see S6A Fig for all DE analysis tools). Using the whole dataset confirms the robustness of this analysis (S6B Fig.).

In terms of experimental design, to get a low $\pi_0$ an experiment would have to change the expression of most genes under study substantially. A significant source of such experiments would be comparisons of different cancer cell lines/tissues, where $\pi_0 = 0.4$ could be considered a reasonable outcome [31]. We, therefore compared the $\pi_0$-s coming from GEO HT-seq

submissions related to search terms "neoplasms" or "cancer" (for the exact query string, please see Methods) with all other non-cancer submissions. There is very little difference in the means and standard deviations of $\pi_0$-s for cancer and non-cancer experiments (0.58 (0.22) and 0.59 (0.24), respectively) (Fig 3C). Filtering by studies mentioning transcription factor led to very similar results (Fig 3D), suggesting that the wide dispersion of $\pi_0$-s is not caused by intentional experimental designs. Also, studies involving cancer and TFs resulted in anti-conservative or uniform $p$-value distributions with similar probabilities to non-cancer/non-TF studies (risk ratios with 95% CI are 0.95 (0.83; 1.07) and 0.99 (0.74; 1.28), respectively).

In addition, controlling for time, taxon, or sequencing platform, did not substantially change the association of DE analysis tools with the $\pi_0$-s (S6C–S6F Fig). Recalculating the $\pi_0$-s with a global FDR algorithm [32] did not lead to substantially different $\pi_0$ distribution, although there appears to be a slight directional shift of the distribution, the mean $\pi_0$ is shifted from 0.58 in the local FDR method to 0.54 in the global FDR method (S7 Fig.). As there is a strong association between both $\pi_0$ and the proportion of anti-conservative $p$-value histograms with the DE analysis tool (Figs 2 and 3), we further checked for and failed to see similar associations with variables from raw sequence metadata, such as the sequencing platform, library preparation strategies, library sequencing strategies, library selection strategies, and library layout (single or paired) (S8–S15 Fig). These results support the conjecture of specificity of the associations with DE analysis tools.

## The sample size distribution of DE-HT experiments indicates persistently low power

We assigned sample sizes to 2,393 GEO submissions (see Materials for details). From these, 91% had sample sizes of 4 or less, 25% had N of just 2 and 12% used a single sample from which to calculate $p$-values, thereby entirely ignoring biological variation (Fig 4A). Only 1% of experiments had sample sizes over 10.

Are the observed sample sizes sufficient to convey reasonable power to DE RNA-seq experiments? Simulations on idealized data with natural variation and effect sizes lead $N = 2$ experiments to 30% to 60% power and $N = 3$ experiments to 50% to 70% power for human/mouse

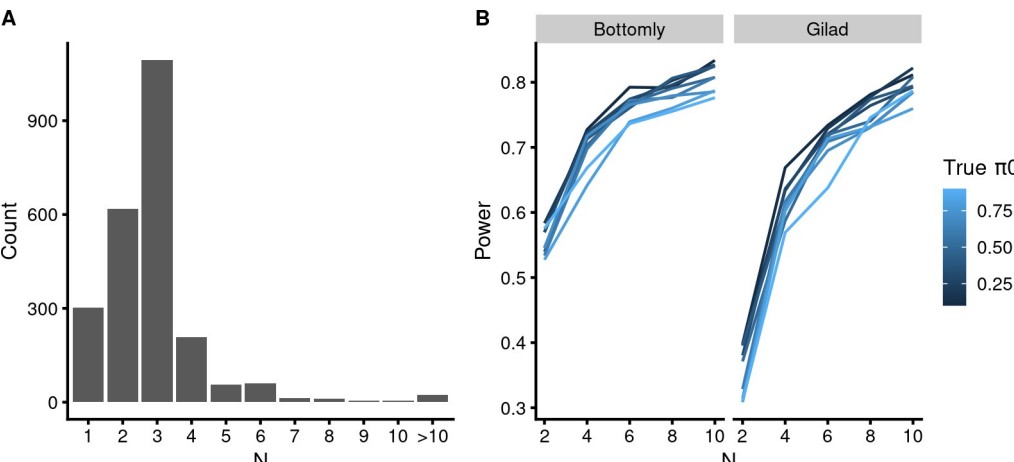

**Fig 4. The sample sizes of HT-seq DE experiments indicate low power.** (**A**) Histogram of 2,393 sample sizes. The data file is in S7 Data. (**B**) Statistical power simulations using different $\pi_0$ settings (shown as shades of blue coloring) and 2 different biological variation settings ("Gilad" corresponds to human liver samples and "Bottomly" to inbred mice; [33]). The data file is in S8 Data.

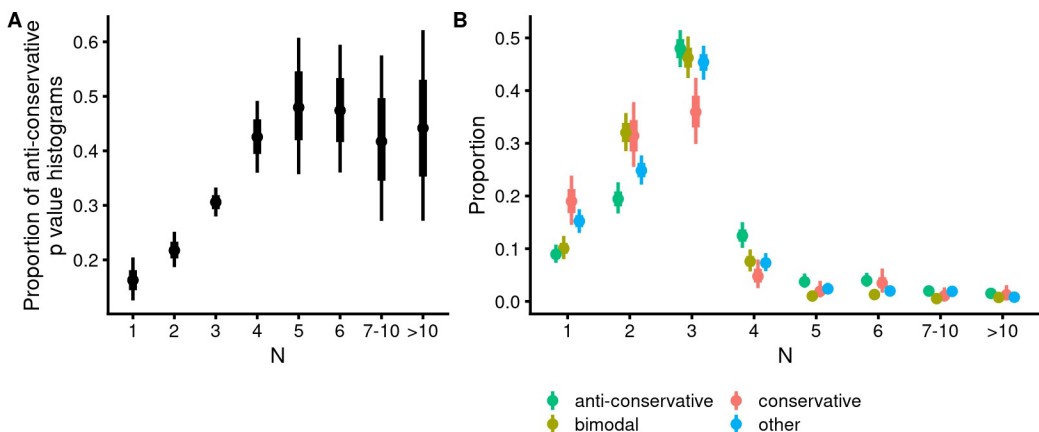

**Fig 5. Association of the sample size of the HT-seq DE experiment with *p*-value distributional class.** (**A**) Increasing sample sizes are associated with an increased fraction of anti-conservative *p*-value distributions. Logistic model [anticons ~ N], Bernoulli likelihood, *N* = 2,392. The data file is in S9 Data. (**B**) Non-anti-conservative *p*-value distributional classes are more likely to have extremely low sample sizes of 1 or 2 and less likely to have sample sizes of 4 or more. The y-axis denotes the proportion of experiments with a given sample size calculated from the total number of experiments in a given *p*-value distributional class. Logistic model [n | trials(total_in_class) ~ Class + Nb + Class:Nb], binomial likelihood, *N* = 2,392. Points denote best estimates for the mean and thick and thin lines denote 66% and 95% credible intervals, respectively. The data file is in S10 Data. The model object related to panel A can be downloaded from https://gin.g-node. org/tpall/geo-htseq-paper/src/v0.1/models/anticons__N.rds. The model object related to panel B can be downloaded from https://gin.g-node.org/tpall/geo-htseq-paper/src/v0.1/models/n%20%7c%20trials%28nn%29__Class%20+%20Nb%20+%20Class:Nb.rds.

experiments, where higher $\pi_0$-s are associated with lower power (Fig 4B). The power of real-life experiments is likely to be even lower, due to imperfect match between the data and the statistical model used to analyze it [34]. Thus, assuming that realistic levels of biological variation are captured in the samples, most DE RNA-seq experiments conducted in eukaryotic systems must be underpowered. This conclusion is consistent with the methodological literature (see Discussion).

## A low sample size lowers the probability of obtaining anti-conservative *p*-value distributions

There is a clear trend, shown in Fig 5A, whereby very small samples of 5 or fewer lead to progressively smaller fractions of anti-conservative *p*-value distributions. While the probability of obtaining an anti-conservative *p*-value distribution is only around 0.1, if the sample size is one, it increases to 0.3 for sample size 3 and further to about 0.5 for sample size 5. We could detect no further increase in this probability as sample size grew further (but note that the vast majority of experiments have sample sizes of 4 or less). As our simulations indicate that reasonably powered (power is around 0.7 to 0.8) experiments start around sample sizes 4 to 5 for multicellular systems, this result suggests that the existing DE analysis algorithms may work poorly on data obtained from underpowered experiments.

Experiments that did not result in anti-conservative *p*-value distributions were less likely to have samples larger than 3 and more likely to have sample sizes of 2 and 1 than the experiments that led to anti-conservative *p*-value distributions (Fig 5B). This effect was especially pronounced for experiments with conservative *p*-value distributions, of which 52% had sample sizes of 1 or 2, as compared with 26% for anti-conservative *p*-value distributions. Overall, the experiments resulting in anti-conservative *p*-value distributions have a slightly larger mean sample size (3.5 versus 2.7; $p = 10^{-9}$).

These results indicate a clear causal link between sample size and technical quality of most current workflows of DE analysis. Thus, a low power not only leads to missed discoveries and overestimated DE-s, as predicted by theory [35], but it also seems to be destructive towards the algorithms that calculate these DE-s.

## Lack of association of sample size with the proportion of true nulls

According to the statistical theory, there is a linear dependence of the power of the experiment on the square root of sample size. An underpowered experiment results in a smaller number of near-zero $p$-values, leading to overestimation of the $\pi_0$ at small N-s, especially at $N = 2$, as confirmed by simulation (Fig 6A). Thus, reducing the power will ultimately convert an anti-conservative $p$-value distribution into a uniform one. However, we not only have very few uniform $p$-value distributions in our empirical data but also few near-one estimated $\pi_0$-s that could be construed as manifestations of low power (Fig 3A). We looked into this further by plotting the $\pi_0$ values versus sample size (Fig 6B). As a result, instead of the expected decrease of the mean $\pi_0$ upon increasing the sample size, we see a slight increase of $\pi_0$-s at sample sizes >5, while the overall correlation is negligible (r = 0.06; 95% CI [0.007, 0.12]). Thus, there does not appear to be a sample size-dependent bias in $\pi_0$-s, as estimated from the GEO-submitted $p$-value distributions. As the true power of most small sample DE RNA-seq experiments is expected to be low, this result leads us to question further the statistical adequacy of the underlying $p$-values, of which the $\pi_0$-s are but single-number summaries. These results show that statistical methods used in HT-seq DE analysis tend to produce highly suspect output even when the $p$-value distribution is anti-conservative.

## Curing of $p$-value histograms by removing low-count features

We observed a slight reduction in the mean number of $p$-values per GEO experiment of anti-conservative and uniform histograms compared to other $p$-value histogram classes, suggesting that $p$-value sets with anti-conservative and uniform shapes are more likely to have been pre-filtered or have been more extensively pre-filtered (S1 Fig). Accordingly, we speculated that by further filtering out features with low counts, we could convert some of the untoward $p$-value

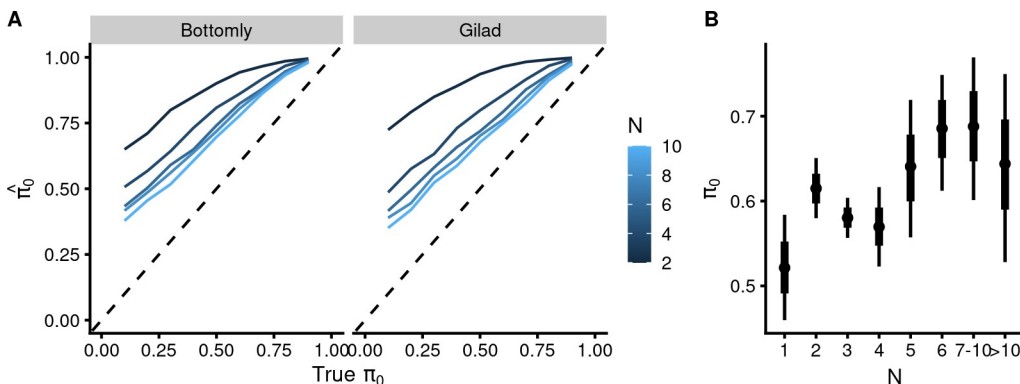

**Fig 6. $\pi_0$-s calculated from anti-conservative p-value sets do not behave in accordance to statistical theory. (A)** Calculated $\pi_0$-s (on the y-axis) from simulated data vs. given "true" proportions of DE-features (on the x-axes). Sample sizes are indicated in color code. The dotted line shows the perfect correspondence between the given $\pi_0$-s and the estimated $\pi_0$. The data file is in S11 Data. **(B)** Dependence of mean $\pi_0$ from the binned sample sizes with 95% CI. Robust linear model [pi0 ~ N], Student's likelihood. Points denote best estimates for the mean and thick and thin lines denote 66% and 95% credible intervals, respectively. The data file is in S12 Data. Model object related to panel B can be downloaded from https://gin.g-node.org/tpall/geo-htseq-paper/src/v0.1/models/pi0%20~%20N.rds.

histograms into anti-conservative or uniform types. Our goal was not to provide optimal interventions for individual datasets, which would require tailoring the filtering algorithm for the requirements of a particular experiment. We merely aim to provide proof of principle that by a simple filtering approach, we can increase the proportion of anti-conservative $p$-value sets and/or reduce the dependence of results on the analysis platform. Therefore, we applied filtering to 3,426 $p$-value sets where we could identify gene expression values (see Methods for details on selecting filtering thresholds).

We found that overall we could increase the proportion of anti-conservative $p$-value histograms by 2.4-fold, from 844 (24.6%) to 2,022 (59.0%), and the number of uniform histograms by 2.5-fold from 8 (0.23%) to 20 (0.6%) (Fig 7A). For all analysis platforms, most rescued $p$-value distributions came from classes "bimodal" and "other," while almost no rescue was detected from conservative histograms (Fig 7B–7F). After removing low count features, the proportion of anti-conservative $p$-value histograms increased for all analysis platforms, with the largest effects observed for cuffdiff and deseq, which presented the lowest pre-rescue fractions of anti-conservative $p$-value histograms (Fig 7G and 7H). Nevertheless, substantial differences between analysis platforms remain, indicating that the removal of low-count features, while generally beneficial, was insufficient to entirely remove the sources of bias originating from the analysis platform. Also, the $\pi_0$-s calculated from the rescued anti-conservative $p$-value sets have very similar distributions compared to $\pi_0$-s from the pre-rescue anti-conservative $p$-value sets and concomitantly very similar dependence on the analysis platform (Fig 7I and 7J).

## Discussion

In this work, we assess the quality of the differential expression analysis by HT-seq based on a large, unbiased NCBI GEO dataset. Our goal was to study real-world statistical inferences made by working scientists. Thus, we study how experimental design choices and analytic decisions of scientists affect the quality of their statistical inferences, with the understanding that in a field where each experiment encompasses ca. 20,000 parallel measurements of DE on average, the quality of statistical inference is highly relevant to the quality of the scientific inference. To the best of our knowledge, this is the first large-scale study to offer quantitative insight into the general quality of experimentation and data analysis of a large field of biomedical science.

We show that

i.  Overall, three-quarters of HT-seq DE experiments result in $p$-value distributions that indicate that assumptions behind the DE tests have not been met. However, for many experiments, a simple exclusion of low-count features rescues the $p$-value distribution.

ii. Very few experiments result in uniform $p$-value distributions that indicate <100 DE features.

iii. The sample sizes of the vast majority of HT-seq DE experiments are inconsistent with reasonably high statistical power to detect true effects.

iv. Nevertheless, the distribution of $\pi_0$-s, the fraction of true null hypotheses in an experiment, which is estimated solely from the $p$-value distributions of experiments presenting anti-conservative $p$-value distributions, peaks at around 0.5, as if in many experiments, most genes were DE. Furthermore, the estimated $\pi_0$-s do not correlate with the sample sizes of the experiments, indicating that the underlying $p$-value sets are problematic for controlling FDR.

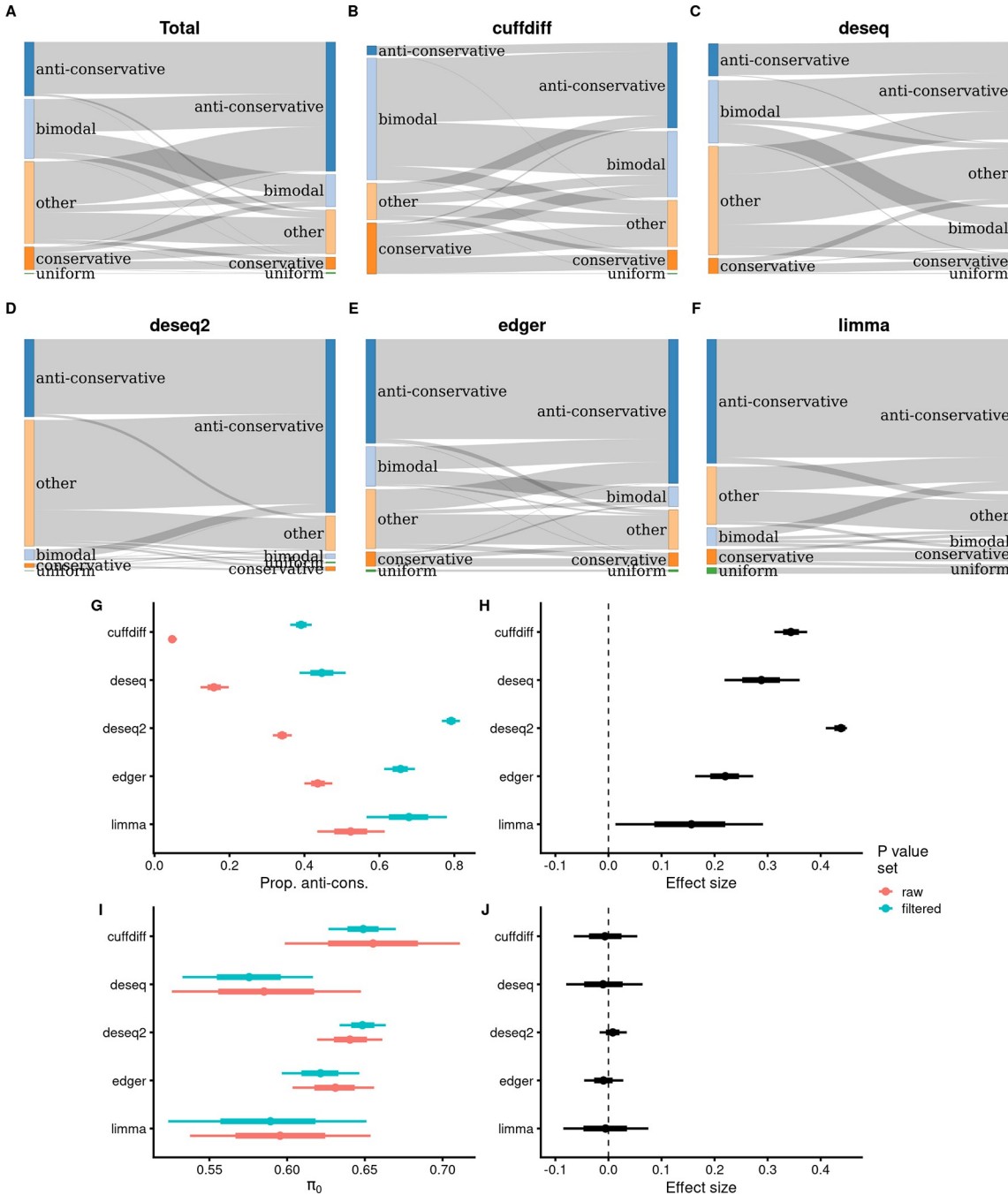

**Fig 7. Removal of low-count features results in an increasing proportion of anti-conservative *p*-value histograms.** (**A-F**) Sankey charts of transformation of *p*-value histogram shape. Ribbon size is linearly proportional to the number of *p*-value sets that change their distributional class. Only the 3,426 experiments that could be subjected to this treatment are depicted. (**A**) Complete data, *N* = 3,426. (**B**) The subset where the *p*-values were calculated with cuffdiff, *N* = 1,116. (**C**) The subset where the *p*-values were calculated with DESeq, *N* = 252. (**D**) The subset where the *p*-values were calculated with DESeq2, *N* = 1,114. (**E**) The subset where the *p*-values were calculated with edgeR, *N* = 515. (**F**) The subset where the *p*-values were calculated with limma, *N* = 73. (**G**) Posterior summaries of anti-conservative *p*-value histogram proportions in raw and filtered *p*-value sets. Filtered *p*-value data is from a Bernoulli model [anticons ~ de_tool], *N* = 3,426. The data files are in S13 Data and in S14 Data (for raw data). (**H**) Effect sizes in percentage points of low-count feature filtering to the proportion of anti-conservative *p*-value histograms. The data files are in S13 Data and in S14 Data (for raw data). (**I**) Posterior summaries of $\pi_0$ values of *p*-value histograms in raw and filtered *p*-value sets. Filtered *p*-value data is the *p*-value from the beta model [pi0 ~ de_tool], *N* = 2,042. The data files are in S15 Data and in S16 Data (for raw data). (**J**) Effect sizes in $\pi_0$ units (percentage points) of low-count feature filtering to $\pi_0$. The data files are in S15 Data and in S16 Data (for raw data). The model object

v. The proportion of anti-conservative *p*-value distributions and the values of $\pi_0$-s are strongly associated with the DE analysis platform.

These results show that not only are most *p*-value sets statistically suspect (as shown by the preponderance of untoward *p*-value distributions), but that even the well-behaved anti-conservative *p*-value sets tend to lead to incompatible statistical inferences with the observed low sample sizes and therefore, low power. Furthermore, the observed association with data analysis platforms of both the *p*-value distributional classes and the $\pi_0$-s of anti-conservative *p*-value sets strongly indicates that DE analysis results in the literature are substantially biased by the analysis platform used.

From our meta-science study, we can conclude that the actual use of statistical tools in the HT-seq DE field is inconsistent with a high quality of statistical inferences. While our results clearly show the pervasiveness of the problem, they by themselves cannot answer the question: how much does it matter downstream for the quality of scientific inferences? However, recent methodological work on individual workflows allows us to provide an answer: it matters a lot. Namely, from simulation studies of existing workflows, it is now becoming clear that all popular *p*-value-based methods that can be used with small samples often fail in the one thing that they were really created for, in FDR control [30,34,36]. Our study sees the results of this widespread failure to control FDR in the high prevalence of low $\pi_0$ estimates, whose lack of dependence on sample size defies the logic of statistical inference, thereby confirming that the poor FDR control observed in simulations is massively carried over into results of actual HT-seq DE data analysis. Collectively, our study and the aforementioned simulation studies indicate that the field is largely built on analyzing low-power experiments, which are unlikely to identify actual effects, but still tend to overestimate the effect sizes of both true and false DE-s of any statistically significant results [35], while presenting an unknowable number of false discoveries as statistically significant. It is hard to imagine a worse state of affairs or that this would not substantially affect the quality of relevant scientific conclusions.

In the following, we will further discuss specific aspects of our work.

## The meaning of the *p*-value distribution

The first thing to notice with the *p*-value distributional classes is that there are extremely few uniform *p*-value distributions, suggesting relatively few actual effects. This unexpected result is made even more surprising by the low statistical power (<40%) of most real-world RNA-seq DE experiments, which should increase the likelihood of encountering uniform *p*-value distributions [37]. As a technical comment, it should be noted that the class assigned by us on a *p*-value histogram depends on an arbitrarily set bin size. Our use of 40 bins leads to histograms, where an experiment with up to around 100 true effects out of 20,000 features could well lead to a uniform histogram because of swamping of the lowermost bin ($p < 0.025$) with *p*-values emanating from true null effects (S17 Fig). If the *p*-values had been calculated correctly, then the lack of uniform *p*-value distributions would indicate that (i) there are almost no experiments submitted to GEO, where the experimental treatment led to only a few DE genes and (ii) that the actual power of GEO-submitted experiments is not low at all. We find both possibilities hard to believe.

While there is a positive temporal trend for the increasing fraction of anti-conservative *p*-value sets, overall, a substantial majority of them fall into shapes indicating that the

assumptions behind the statistical tests, which produced these $p$-values, have not been met. Expressly, $p$-value distributions that are not anti-conservative/uniform relinquish statistical control of the experiment over type I errors, on the presence of which later $p$-value adjustments are predicated [21]. Consequently, such $p$-value sets have effectively relinquished their statistical meaning and are thus problematic for scientific interpretation and further analysis, including FDR/q value calculations. Specifically, conservative $p$-value histograms are thought to be caused by small samples, indicative of reduced power and overly conservative control of FDR [30]. The conjecture of low power is supported by our data in that experiments with conservative $p$-value distributions are the most likely to have sample sizes of 1 or 2 (Fig 5B).

In contrast, anti-conservative $p$-value distributions that have more than expected small $p$-values, and thus led to reduced $\pi_0$ estimates in our analysis, tend to lead to loss of FDR control [30]. In general, even a small deviation from the uniformity of the $p$-value distribution of true nulls is expected to throw off the overall FDR control, which, alas, is the main reason for using these methods for the statistical analysis of DE [30]. In fact, without FDR control, any statistical method that converts continuous effects into statistical significance through binary decisions can do more harm than good [11].

Currently, most DE analysis platforms use parametric tests accompanied by GLM regression. However, it has been recently shown for the often-used parametric tests that their FDR control can be exceedingly poor, while the nonparametric Wilcoxon rank sum test achieves better results [34]. Furthermore, it is becoming clear that data normalization can introduce bias into HT-seq DE analysis, which cannot be corrected by many of the currently widely used methods [38,39]. Indeed, widely used DE analysis tools use RNA-seq data preprocessing strategies, which are vulnerable to situations where a significant fraction of features change expression, and different samples exhibit differing total RNA concentrations [25,40].

In general, the analysis tools used for differential expression testing allow for a plethora of choices, including distributional models, data transformations, and basic analytic strategies, which can lead to different results through different trade-offs [41–43]. Our finding of a high proportion of unruly $p$-value distributions suggests that the wide variety of individual data preprocessing/DE testing workflows, in their actual use by working scientists, results in an overall poor outcome. Although this situation has been steadily improving over the last decade, at least for the limma, edgeR, and DESeq2 users, there remains substantial room for further improvements.

## Possible causes for analysis platform-driven bias

What could cause the systematic differences between analysis platforms that we see in $p$-value distributions and $\pi_0$ values? The significant points of divergence in the analytic pathway include raw sequence preprocessing, aligning sequences to a reference sequence, counts normalization, and differential expression testing [42,44]. Our inability to abolish analysis platform-driven bias by removal of low-count features suggests that the pre-filtering of data is not a major source of this bias. Although in principle, any tool can be used in many workflows, different tools tend to offer their users different suggested workflows and different amounts of automation for data preprocessing and analysis. For example, Cuffdiff almost fully automates the analysis, including the removal of biases [45], DESeq2 workflow suggests some preprocessing steps to speed-up computations but uses automation to remove biases from input data [46], whereas edgeR [47] and limma-voom require more interactive execution of separate preprocessing and analysis steps [48,49]. We speculate that the popularity of Cuffdiff and DESeq2 partly lies in their automation, as the user is largely relieved from decision-making. However, we found that cuffdiff is associated with the smallest proportion of anti-conservative $p$-value

histograms, whereas limma and edger, with their more hands-on approach, are associated with the highest proportions of anti-conservative histograms. Interestingly, limma and edgeR use very different distributional models for DE testing, supporting the notion that it might be data preprocessing rather than the statistical test that has the most impact on the success of DE analysis [25]. However, limma and edgeR were not the top performers on the $\pi_0$-metric, where the highest performance is associated with Cuffdiff, whose ability to provide anti-conservative $p$-value distributions was the least impressive (note that the $\pi_0$-s were calculated only from the experiments with well-distributed $p$-values).

## Power considerations

The prevalent sample sizes of 2 to 3 in published HT-seq DE experiments are incompatible with our simulations and the literature in terms of providing enough power to the experiment to reliably find most DE genes. For example, with the currently favored sample size of 3, for most genes in isogenic yeast, effect sizes of at least 4-fold seem to be required for successful analysis, and the overall minimal acceptable sample size can be from 4 to 6 [37,50]. In animals, a reasonable sample size is expected to be substantially larger, over 10 for most genes, which are not highly expressed, and for cancer samples, it seems to be well over 20 [37,43,51–53]. This raises the possibility that nearly all experiments, which should have resulted in uniform distributions, were somehow shifted into other distributional classes. The low power to detect DE of most genes also means that excluding most low-count genes (which means most genes) should be encouraged because it would increase the power to detect the highly expressed genes, while DE of lower-expressed genes would have been missed anyway. Of course, when the scientific goal is to study the DE of genes that are not highly expressed, there is no real alternative to substantially increasing the sample size.

## Estimated $\pi_0$-s suggest experimental design problems

The DE analysis tool-specific means of $\pi_0$ values range from 0.5 (tool "unknown") to 0.7 (tool "cuffdiff"), showing that by this criterion in an average RNA-seq experiment about half of the cellular RNA-s are expected to change their expression levels (Fig 3B). In principle, a low $\pi_0$ could reflect true differential expression as a causal response to experimental treatment, or it could be an artefact of suboptimal data analysis. To further investigate this, we separately analyzed 2 cases of studies related either to cancer or transcription factors, assuming that available $p$-value sets reflect the situation where DE is profiled in cancer cells/tissues or after transcription factor interrogation, respectively. Due to large-scale rearrangements in cancer genomes and the ability of many TFs to change the expression of many genes, these studies are expected to lead to lower $\pi_0$-s, perhaps in the 0.4 range [31]. However, in both cases, we saw essentially unchanging $\pi_0$ distributions, which suggests that the low $\pi_0$-s in our dataset are indicative of problematic analytic workflows. This conclusion is further strengthened by the observed lack of negative correlations between $\pi_0$ and sample size, which is a proxy of statistical power.

It should be added that as most data normalization workflows assume that most genes are not DE (and that total RNA concentration are stable across experimental conditions), any study that results in a low $\pi_0$ should strive to explicitly address this fact in the experimental setup, data analysis, and interpretation of results. It has been argued that most genome-wide DE studies ever conducted, including by HT-seq, have used experimental designs that would make it impossible to untangle such global effects, at least in a quantitatively accurate way [54,55]. The issue lies in the use of internal standards in normalizing the counts for different RNA-s, which leads to wrong interpretations if most genes change expression in 1 direction. To overcome this problem, one could use spike-in RNA standards and compositional

normalization [40]. However, spike-in normalization requires great care to work correctly in such extreme cases [39,56], and it is rarely used outside single-cell RNA-seq [40]. Thus, it seems likely that many or most of the low $\pi_0$ experiments represent technical failures, most likely during data normalization.

## Considerations for improvement of best practices

Currently, at least 29 different DE RNA-seq analysis platforms exist, which, among other differences, use 12 different distributional models for DE testing [50], while it is becoming apparent that all popular distributional models have trouble achieving promised FDR levels [34]. At a minimum, a winnowing of recommended analytic choices is needed. Nevertheless, design and analysis of a specific DE RNA-seq experiment would have to be responsive to several factors, including (i) the study organism and its genetic background, pertaining to biological variation and thus sample size; (ii) the expected $\pi_0$, pertaining to experimental design, like the use of spike-in probes, and data analysis; (iii) the number of genes of interest, pertaining to sample size through the multiple testing problem; (iv) the expected expression levels and biological variation of genes of interest, and whether DE of gene isoforms is of interest, pertaining to sample size and to sequencing depth; and (v) the structure of experiment pertaining to analysis via GLM (is it multi-group, multi-center, multi-experimenter?). As no analytic workflow has been shown to outperform the others systematically [25,41–43], foolproof best practices are still out of reach.

Another class of classic nonparametric *p*-value calculation methods, of which the Wilcoxon rank-sum test is the most popular, is free of distributional assumptions and has good control of FDR, but they need sample sizes of at least 10 to achieve good power [34]. We note that such sample sizes are scarce in the HT-seq DE analysis field. Recently, *p*-value-free approaches to FDR control that do not depend on distributional models have been proposed that look promising even in small samples [36,57]. These methods would, of course, transcend any problems caused by *p*-values (as well as the diagnostic value of the p distribution), except for the more general malady of effect size overestimation at low power, which comes along with any method that dichotomizes continuous data into "discoveries" and the rest [11].

While our results leave no doubt about the pervasiveness of problems in the HT-seq DE field, being meta-scientific by nature, they can offer relatively little in terms of suggestions for specific improvements to various workflows used in the field. Although examining *p*-value histograms and $\pi_0$-s derived from them and more stringently excluding low-count features should make their way into every HT-seq DE analysis, we have no reason to think that these steps alone will cure the field of its ailments. Also, while removing low-expressed genes before model fitting in DE analysis can have a substantial positive effect on the sensitivity of detection of differentially expressed genes [58,59], this comes with the cost of excluding about half the genes from analysis as lowly expressed [60]. While there is no single best way to exclude features from analysis, an adaptive filtering procedure has been recently proposed [61]. In deciding, how many low-count features to exclude in a given study, both scientific and statistical considerations should play a part. Namely, quantification of differential expression of low-expressed genes appears to be unreliable by existing DE analysis tools [62], and the sample size needed for accurate measurement of DE has a strong inverse relationship with the expression level of the gene [43].

Increasing the power by increasing the sample sizes while designing the experiments to capture the full extent of relevant biological variation in those samples would be another obvious, if expensive, recommendation. In terms of DE analysis, a significant danger seems to be overly reducing the within-data variation during data preprocessing so that biological variation that may be present in the raw data is lost during the analysis.

In conclusion, we do not think there is an apparent single fix to the problems afflicting a large field with quite heterogeneous scientific questions and corresponding experimental designs, and it may well be that the methodology that works for HT-seq DE is still in the future. We do not think it impossible that fixing the RNA-seq field requires, as a first step, the development of cheaper ways for conducting experiments, which would then make well-powered experiments practically feasible, which in turn could lead to the development of analytic workflows that work.

## Methods

### NCBI GEO database query and supplementary files

NCBI GEO database queries were performed using Bio.Entrez Python package and by sending requests to NCBI Entrez public API. The exact query string to retrieve GEO HT-seq datasets was "expression profiling by high-throughput sequencing[DataSet Type] AND ("2000-01-01" [PDAT]: "2020-12-31" [PDAT])." Accession numbers of cancer-related datasets were identified by amending the original query string with "AND ("neoplasms" [MeSH Terms] OR cancer[All Fields])." FTP links from GEO datasets document summaries were used to download supplementary file names. Supplementary file names were filtered for downloading, based on file extensions, to keep file names with "tab," "xlsx," "diff," "tsv," "xls," "csv," "txt," "rtf," and "tar" file extensions. We dropped the file names where we did not expect to find $p$-values using the regular expression "series_matrix\.txt\.gz$|filelist\.txt$|readme|\.bam(\.tdf|$)|\.bai(\.gz|$)| \.sam(\.gz|$)|\.csfasta|\.fa(sta)?(\.gz|\.bz2|\.txt\.gz|$)|\.f(a|n)a(\.gz|$)|\.wig|\.big[Ww]ig$|\.bw (\.|$)|\.bed([Gg]raph)?(\.tdf|\.gz|\.bz2|\.txt\.gz|$)|(broad_)?lincs|\.tdf$|\.hic$|\.rds(\.gz|$)|\.tar\. gz$|\.mtx(\.gz$|$)|dge\.txt\.gz$|umis?\.txt\.gz$."

### NCBI supplementary file processing

Downloaded files were imported using the Python pandas package and searched for unadjusted $p$-value sets. Unadjusted $p$-value sets and summarized expression level of associated genomic features were identified using column names. $P$-value columns from imported tables were identified by regular expression "p[^a-zA-Z]{0,4}val," from these, adjusted $p$-value sets were identified using the regular expression "adj|fdr|corr|thresh" and omitted from further analysis. We algorithmically tested the quality of identified $p$-value sets and removed from further analysis apparently truncated or right-skewed sets, $p$-value sets that were not in the 0 to 1 range, and $p$-value sets that consisted entirely of NaN values. Columns with expression levels of genomic features were identified by using the following regular expressions: "basemean," "value," "fpkm," "logcpm," "rpkm," "aveexpr." Where expression level data were present, raw $p$-values were further filtered to remove low-expression features using the following thresholds: basemean = 10, logcpm = 1, rpkm = 1, fpkm = 1, aveexpr = 3.32. Basemean is a mean of library-size normalized counts of all samples, logcpm is the mean log2 counts per million, rpkm/fpkm is reads/fragments per kilobase of transcript length per million reads, aveexpr is an average expression across all samples, in log2 CPM units, whereas CPM is counts per million. Row means were calculated when there were multiple expression level columns (e.g., for each contrast or sample) in the table. Filtered $p$-value sets were stored and analyzed separately.

### Classification of $p$-value histograms

Raw $p$-value sets were classified based on their histogram shape. The histogram shape was determined based on the presence and location of peaks. $P$-value histogram peaks (bins) were detected using a quality control threshold described in [27], a Bonferroni-corrected alpha-level

quantile of the cumulative function of the binomial distribution with size m and probability p. Histograms, where none of the bins was over QC-threshold, were classified as "uniform." Histograms, where bins over the QC threshold started either from the left or right boundary and did not exceed 1/3 of the 0 to 1 range, were classified as "anti-conservative" or "conservative," respectively. Histograms with peaks or bumps in the middle or with non-continuous left- or right-side peaks were classified as "other." Finally, histograms with left- and right-side peaks were classified as "bimodal."

## Calculation of $\pi_0$ statistic

Raw $p$-value sets with an anti-conservative shape were used to calculate the $\pi_0$ statistic. The $\pi_0$ statistic was calculated using the local FDR method implemented in limma::PropTrueNullBy-LocalFDR [48] and, independently, Storey's global FDR smoother method [32] as implemented in gdsctools [63] Python package. Differential expression analysis tools info was collected from different sources—extracted from full-text articles via NCBI PubMed Central API, extracted from GEO summaries, extracted from supplementary file names, extracted from text appended to $p$-value set names, and finally, as auxiliary information, inferred from column names pattern, by using following heuristics, cuffdiff (column name = "fpkm" and "p_value") [45], DESeq/DESeq2 (column name = "basemean" and "pval" or "pvalue", respectively) [46], EdgeR (column name = "logcpm") [47], and limma (column name = "aveexpr," "p.value," and PDAT > 2014-01-01) [48], all sets that remained unidentified were binned as "unknown." We used following regular expression to extract DE analysis tool names from lower-case converted text "deseq2?|de(g|x)seq|rockhopper|cuff(diff|links)|edger|clc(bio)?? genomics|igeak|bayseq|samseq|noiseq|ebseq|limma|voom|sleuth|partek|(nrsa|nascent rna seq)|median ratio norm|rmats|ballgown|biojupie|seurat|exdega".

## Sample size determination

Sample sizes were algorithmically determined for tables containing a single column of $p$-values by dividing the number of samples with the number of $p$-value sets + 1. Only balanced samples were retained, and all 4 or more sample sizes were manually verified.

## Modeling

Bayesian modeling was done using R libraries rstan vers. 2.21.3 [64] and brms vers. 2.16.3 [65]. Models were specified using extended R lme4 [66] formula syntax implemented in the R brms package. We used weak priors to fit models. We run minimally 2,000 iterations and 3 chains to fit models. When suggested by brms, Stan NUTS control parameter adapt_delta was increased to 0.95–0.99 and max_treedepth to 12–15.

## RNA-seq simulation

RNA-seq experiment simulation was done with polyester R package [67], and differential expression was assessed using DESeq2 R package [46] using default settings. Code and workflow used to run and analyze RNA-seq simulations are deposited in Zenodo with doi: 10.5281/zenodo.4463804 (https://doi.org/10.5281/zenodo.4463804). RNA-seq simulations to determine relationship between sample size, $\pi_0$ and power were done using PROPER package [33].

## Code and raw data

The code to produce the raw dataset is available as a snakemake workflow [68] on the rstats-tartu/geo-htseq Github repo (https://github.com/rstats-tartu/geo-htseq). The raw dataset

produced by the workflow is deposited in Zenodo https://zenodo.org with doi: 10.5281/zenodo.7529832 (https://doi.org/10.5281/zenodo.7529832). The code to produce the article's figures and models is deposited on the rstats-tartu/geo-htseq-paper Github repo (https://github.com/rstats-tartu/geo-htseq-paper). Individual model objects are deposited in G-Node with doi: 10.12751/g-node.p34qyd (https://doi.org/10.12751/g-node.p34qyd).

ggplot2 vers. 3.3.1. [69] R library was used for graphics. Data wrangling was done using tools from the tidyverse package [70]. Bayesian models were converted to tidy format and visualized using the tidybayes R package [71].

## Supporting information

**S1 Fig. Reduced number of features in anti-conservative and uniform p-value sets.** (A) *P*-value set size distribution. Dashed line denotes the median number of features. From each GEO series, only 1 random set was considered, $N = 4,616$ *p*-value sets. The data file is in S17 Data. (B) Robust linear modeling of number of features in anti-conservative and uniform vs. non-anti-conservative *p*-value sets [log10_n_pvalues ~ anticons, Student's t likelihood], $N = 4,616$. Points denote best fit of linear model. Thick and thin lines denote 66% and 95% credible region, respectively. The data file is in S18 Data. The model object related to panel B can be downloaded from https://gin.g-node.org/tpall/geo-htseq-paper/src/v0.1/models/log10_n_pvalues%20~%20anticons.rds.
(TIFF)

**S2 Fig. The increasing proportion of anti-conservative histograms.** Bernoulli model [anticons ~ year], $N = 4,616$. Lines denote best fit of linear model. Shaded area denotes 95% credible region. The data file is in S19 Data. The model object related to figure can be downloaded from https://gin.g-node.org/tpall/geo-htseq-paper/src/v0.1/models/anticons_year.rds.
(TIFF)

**S3 Fig. Association of the p-value histogram class with a differential expression analysis tool.** (A) Time courses for proportions of different *p*-value histogram classes for the 9 most frequent DE analysis platforms. Lines denote best fit of the model [Class ~ year + (year | de_tool), categorical likelihood]. Shaded areas denote 95% credible regions. $N = 4,616$. The data file is in S20 Data. (B) Association of *p*-value histogram type with DE analysis tool; data is restricted to 2018–2020 GEO submissions. Points denote best fit of the model [n | trials(total_in_de_tool) ~ Class + de_tool + Class:de_tool, binomial likelihood]. Thick and thin lines denote 66% and 95% credible intervals, respectively. $N = 2,930$. The data file is in S21 Data. The model object related to panel A can be downloaded from https://gin.g-node.org/tpall/geo-htseq-paper/src/v0.1/models/Class_year__year_detool_year.rds. The model object related to panel B can be downloaded from https://gin.g-node.org/tpall/geo-htseq-paper/src/v0.2/models/n__trials%28total_in_de_tool%29__Class_de_tool_Class:de_tool_2018up.rds.
(TIFF)

**S4 Fig. No single differential expression analysis tool dominates the field.** Y-axis shows the proportion of analysis platforms, x-axis shows publication year of GEO submission, $N = 4,616$. The data file is in S22 Data.
(TIFF)

**S5 Fig. DE analysis tool conditional effects from binomial logistic models for proportion of anti-conservative p value histograms.** (A) Simple model [anticons ~ de_tool], $N = 4,616$. The data file is in S23 Data. (B) Simple model [anticons ~ de_tool] fitted on complete data, $N = 14,813$. The data file is in S24 Data. (C) Model conditioned on year of GEO submission

[anticons ~ year + de_tool], $N = 4,616$. The data file is in S25 Data. (D) Model conditioned on studied organism (human/mouse/other) [anticons ~ organism + de_tool], $N = 3,886$. The data file is in S26 Data. (E) Varying intercept model [anticons ~ de_tool + (1 | model)] where "model" stands for sequencing instrument model, $N = 3,778$. The data file is in S27 Data. (F) Varying intercept and slope model [anticons ~ de_tool + (de_tool | model)], $N = 3,778$. The data file is in S27 Data. Points denote best fit of linear model. Thick and thin lines denote 66% and 95% credible interval, respectively. The model object related to panel A can be downloaded from https://gin.g-node.org/tpall/geo-htseq-paper/src/v0.1/models/anticons_detool. rds. The model object related to panel B can be downloaded from https://gin.g-node.org/tpall/ geo-htseq-paper/src/v0.1/models/anticons_detool_all.rds. The model object related to panel C can be downloaded from https://gin.g-node.org/tpall/geo-htseq-paper/src/v0.2/models/ anticons_year_detool.rds. The model object related to panel D can be downloaded from https://gin.g-node.org/tpall/geo-htseq-paper/src/v0.2/models/anticons_organism_detool.rds. The model object related to panel E can be downloaded from https://gin.g-node.org/tpall/geo-htseq-paper/src/v0.1/models/anticons_detool__1_model.rds. The model object related to panel F can be downloaded from https://gin.g-node.org/tpall/geo-htseq-paper/src/v0.1/ models/anticons_detool__detool_model.rds.
(TIFF)

**S6 Fig. DE analysis tool-conditional effects from beta regression modeling of $\pi_0$.** (A) Simple model [pi0 ~ de_tool] fitted on sample, $N = 1,188$. The data file is in S4 Data. (B) Simple model [pi0 ~ de_tool] fitted on complete data, $N = 3,898$. The data file is in S28 Data. (C) Model conditioned on year of GEO submission [pi0 ~ year + de_tool], $N = 1,188$. The data file is in S29 Data. (D) Model conditioned on studied organism (human/mouse/other) [pi0 ~ organism + de_tool], $N = 993$. The data file is in S30 Data. (E) Varying intercept model [pi0 ~ de_tool + (1 | model)] where "model" stands for sequencing instrument model, $N = 959$. The data file is in S31 Data. (F) Varying intercept/slope model [pi0 ~ de_tool + (de_tool | model)], $N = 959$. The data file is in S31 Data. Points denote best fit of linear model. Thick and thin lines denote 66% and 95% credible interval, respectively. The model object related to panel A can be downloaded from https://gin.g-node.org/tpall/geo-htseq-paper/src/v0.1/models/pi0_detool_sample. rds. The model object related to panel B can be downloaded from https://gin.g-node.org/tpall/ geo-htseq-paper/src/v0.1/models/pi0_detool_full_data.rds. The model object related to panel C can be downloaded from https://gin.g-node.org/tpall/geo-htseq-paper/src/v0.2/models/pi0_ year_detool.rds. The model object related to panel D can be downloaded from https://gin.g-node.org/tpall/geo-htseq-paper/src/v0.2/models/pi0_organism_detool.rds. The model object related to panel E can be downloaded from https://gin.g-node.org/tpall/geo-htseq-paper/src/ v0.1/models/pi0_detool__1_model.rds. The model object related to panel F can be downloaded from https://gin.g-node.org/tpall/geo-htseq-paper/src/v0.1/models/pi0_detool__ detool_model.rds.
(TIFF)

**S7 Fig. Comparison of π0 values computed by 2 different methods.** Local FDR method is from limma R package function propTrueNull. Smoother method is from q value R package. A, density histogram. B, scatter plot. Dashed line has intercept = 0 and slope = 1. The data file is in S32 Data.
(TIFF)

**S8 Fig. Dependency of $\pi_0$ on sequencing instrument model.** Points denote best fit of linear model ([pi0 ~ model], beta distribution, $N = 959$). Thick and thin lines denote 66% and 95% credible interval, respectively. The data file is in S33 Data. The model object related to figure

can be downloaded from https://gin.g-node.org/tpall/geo-htseq-paper/src/v0.1/models/pi0__
model.rds.
(TIFF)

**S9 Fig. Dependency of $\pi_0$ on library strategy.** Points denote best fit of linear model ([pi0 ~
library_strategy], beta distribution, *N* = 959). Thick and thin lines denote 66% and 95% credible interval, respectively. The data file is in S34 Data. The model object related to figure can be
downloaded from https://gin.g-node.org/tpall/geo-htseq-paper/src/v0.1/models/pi0__
librarystrategy.rds.
(TIFF)

**S10 Fig. Dependency of $\pi_0$ on library selection.** Points denote best fit of linear model ([pi0 ~
library_selection, beta likelihood], *N* = 959). Thick and thin lines denote 66% and 95% credible
interval, respectively. The data file is in S35 Data. The model object related to figure can be
downloaded from https://gin.g-node.org/tpall/geo-htseq-paper/src/v0.1/models/pi0__
libraryselection.rds.
(TIFF)

**S11 Fig. Dependency of $\pi_0$ on library layout.** Points denote best fit of linear model ([pi0 ~
library_layout, beta likelihood], *N* = 959.). Thick and thin lines denote 66% and 95% credible
interval, respectively. The data file is in S36 Data. The model object related to figure can be
downloaded from https://gin.g-node.org/tpall/geo-htseq-paper/src/v0.2/models/pi0__1_
librarylayout.rds.
(TIFF)

**S12 Fig. Dependency of proportion of anti-conservative histograms on sequencing platform.** Points denote best fit of linear model ([anticons ~ model, bernoulli likelihood],
*N* = 3,778). Thick and thin lines denote 66% and 95% credible interval, respectively. The data
file is in S37 Data. The model object related to figure can be downloaded from https://gin.g-
node.org/tpall/geo-htseq-paper/src/v0.2/models/anticons__1_model.rds.
(TIFF)

**S13 Fig. Dependency of proportion of anti-conservative histograms on library strategy.**
Points denote best fit of linear model ([anticons ~ library_strategy, bernoulli likelihood],
*N* = 3,778). Thick and thin lines denote 66% and 95% credible interval, respectively. The data
file is in S38 Data. The model object related to figure can be downloaded from https://gin.g-
node.org/tpall/geo-htseq-paper/src/v0.2/models/anticons__librarystrategy.rds.
(TIFF)

**S14 Fig. Dependency of proportion of anti-conservative histograms on library selection.**
Points denote best fit of linear model ([anticons ~ library_selection, bernoulli likelihood],
*N* = 3,778). Thick and thin lines denote 66% and 95% credible interval, respectively. The data
file is in S39 Data. The model object related to figure can be downloaded from https://gin.g-
node.org/tpall/geo-htseq-paper/src/v0.1/models/anticons__libraryselection.rds.
(TIFF)

**S15 Fig. Dependency of proportion of anti-conservative histograms on library layout.**
Points denote best fit of linear model ([anticons ~ library_layout, bernoulli likelihood],
*N* = 3,778). Thick and thin lines denote 66% and 95% credible interval, respectively. The data
file is in S40 Data. The model object related to figure can be downloaded from https://gin.g-
node.org/tpall/geo-htseq-paper/src/v0.1/models/anticons__librarylayout.rds.
(TIFF)

**S16 Fig. Removal of low-count features results in an increasing proportion of anti-conservative p-value histograms.** (A) Anti-conservative *p*-value histogram proportions in raw and filtered *p*-value sets for DE analysis programs. Raw *p*-value data is the same as in S5A Fig. Filtered *p*-value data is from a simple Bernoulli model [anticons ~ de_tool], $N = 3,426$. The data files are in S13 Data and in S14 Data (for raw data). (B) Effect size of low-count feature filtering to proportion of anti-conservative *p*-values. The data files are in S13 Data and in S14 Data (for raw data). (C) $\pi_0$ estimates for raw and filtered *p*-value sets. Raw *p*-value data is the same as in S6A Fig and filtered *p*-value data is from the beta model [pi0 ~ de_tool], $N = 2,042$. The data files are in S15 Data and in S16 Data (for raw data). (D) Effect size of low-count feature filtering to $\pi_0$. The data files are in S15 Data and in S16 Data (for raw data). Points denote model best fit. Thick and thin lines denote 66% and 95% CIs, respectively.
(TIFF)

**S17 Fig. Simulated RNA-seq data shows that histograms from p-value sets with around 100 true effects out of 20,000 features can be classified as "uniform".** RNA-seq data was simulated with polyester R package on 20,000 transcripts from human transcriptome using grid of 3, 6, and 10 replicates and 100, 200, 400, and 800 effects for 2 groups. Fold changes were set to 0.5 and 2. Differential expression was assessed using DESeq2 R package using default settings and group 1 versus group 2 contrast. Effects denotes in facet labels the number of true effects and N denotes number of replicates. Red line denotes QC threshold used for dividing p histograms into discrete classes. Code and workflow used to run these simulations is available on Github: https://github.com/rstats-tartu/simulate-rnaseq. Raw data of the figure is available on Zenodo https://zenodo.org with doi: 10.5281/zenodo.4463803.
(TIFF)

**S1 Data. Data used to produce Fig 2A.**
(CSV)

**S2 Data. Data used to produce Fig 2B.**
(CSV)

**S3 Data. Data used to produce Fig 3A.**
(CSV)

**S4 Data. Data used to produce Figs 3B and S6A.**
(CSV)

**S5 Data. Data used to produce Fig 3C.**
(CSV)

**S6 Data. Data used to produce Fig 3D.**
(CSV)

**S7 Data. Data used to produce Fig 4A.**
(CSV)

**S8 Data. Data used to produce Fig 4B.**
(CSV)

**S9 Data. Data used to produce Fig 5A.**
(CSV)

**S10 Data. Data used to produce Fig 5B.**
(CSV)

**S11 Data. Data used to produce Fig 6A.**
(CSV)

**S12 Data. Data used to produce Fig 6B.**
(CSV)

**S13 Data. Data used to produce Figs 7G, 7H, S16A, and S16B.**
(CSV)

**S14 Data. Raw data used to produce Figs 7G, 7H, S16A, and S16B.**
(CSV)

**S15 Data. Data used to produce Figs 7I, 7J, S16C, and S16D.**
(CSV)

**S16 Data. Raw data used to produce Figs 7I, 7J, S16C, and S16D.**
(CSV)

**S17 Data. Data used to produce S1A Fig.**
(CSV)

**S18 Data. Data used to produce S1B Fig.**
(CSV)

**S19 Data. Data used to produce S2 Fig.**
(CSV)

**S20 Data. Data used to produce S3A Fig.**
(CSV)

**S21 Data. Data used to produce S3B Fig.**
(CSV)

**S22 Data. Data used to produce S4 Fig.**
(CSV)

**S23 Data. Data used to produce S5A Fig.**
(CSV)

**S24 Data. Data used to produce S5B Fig.**
(CSV)

**S25 Data. Data used to produce S5C Fig.**
(CSV)

**S26 Data. Data used to produce S5D Fig.**
(CSV)

**S27 Data. Data used to produce S5E and S5F Fig.**
(CSV)

**S28 Data. Data used to produce S6B Fig.**
(CSV)

**S29 Data. Data used to produce S6C Fig.**
(CSV)

**S30 Data. Data used to produce S6D Fig.**
(CSV)

**S31 Data. Data used to produce S6E and S6F Fig.**
(CSV)

**S32 Data. Data used to produce S7 Fig.**
(CSV)

**S33 Data. Data used to produce S8 Fig.**
(CSV)

**S34 Data. Data used to produce S9 Fig.**
(CSV)

**S35 Data. Data used to produce S10 Fig.**
(CSV)

**S36 Data. Data used to produce S11 Fig.**
(CSV)

**S37 Data. Data used to produce S12 Fig.**
(CSV)

**S38 Data. Data used to produce S13 Fig.**
(CSV)

**S39 Data. Data used to produce S14 Fig.**
(CSV)

**S40 Data. Data used to produce S15 Fig.**
(CSV)

## Acknowledgments

We are grateful to Toomas Mets (University of Tartu) for critically reading the manuscript and Niilo Kaldalu (University of Tartu) and Margus Pihlak (Tallinn University of Technology) for valuable discussions.

## Author Contributions

**Conceptualization:** Taavi Päll, Ülo Maiväli.

**Data curation:** Taavi Päll.

**Formal analysis:** Taavi Päll, Hannes Luidalepp, Ülo Maiväli.

**Investigation:** Taavi Päll.

**Methodology:** Taavi Päll, Ülo Maiväli.

**Resources:** Tanel Tenson.

**Software:** Taavi Päll.

**Supervision:** Tanel Tenson.

**Visualization:** Taavi Päll, Ülo Maiväli.

**Writing – original draft:** Taavi Päll, Ülo Maiväli.

**Writing – review & editing:** Taavi Päll, Hannes Luidalepp, Tanel Tenson, Ülo Maiväli.

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
