## [Editor Report · Decision Letter 0]

11 Aug 2022

Dear Dr Maiväli, 

Thank you for submitting your manuscript entitled "A field-wide assessment of differential expression profiling by high throughput sequencing reveals widespread bias" for consideration as a Meta-Research Article by PLOS Biology. Please accept my sincere apologies for the delay in getting back to you we consulted with an academic editor about your submission. 

Your manuscript has now been evaluated by the PLOS Biology editorial staff, as well as by an academic editor with relevant expertise, and I am writing to let you know that we would like to send your submission out for external peer review.

Once your full submission is complete, your paper will undergo a series of checks in preparation for peer review. After your manuscript has passed the checks it will be sent out for review. To provide the metadata for your submission, please Login to Editorial Manager (https://www.editorialmanager.com/pbiology) within two working days, i.e. by Aug 13 2022 11:59PM.

Kind regards,

Richard

Richard Hodge, PhD

Associate Editor, PLOS Biology

rhodge@plos.org

PLOS

---

## [Decision Letter · Decision Letter 1]

21 Sep 2022

Dear Dr Maiväli,

Thank you for your patience while your manuscript "A field-wide assessment of differential expression profiling by high throughput sequencing reveals widespread bias" was peer-reviewed at PLOS Biology. Please accept my sincere apologies for the delays that you have experienced during the peer review process. Your manuscript has now been evaluated by the PLOS Biology editors, an Academic Editor with relevant expertise, and by two independent reviewers. 

The reviews are attached below. You will see that the reviewers find your manuscript interesting, but ask for some additional analyses to increase its impact for the field, including an investigation into the cause of the bias in the analyses and ways in which the issue can be mitigated. In addition, they ask for several reporting details to be added, including the filtering steps and DE analysis tools used in the different datasets.

In light of the reviews, we will not be able to accept the current version of the manuscript, but we would welcome re-submission of a much-revised version that takes into account the reviewers' comments. We cannot make any decision about publication until we have seen the revised manuscript and your response to the reviewers' comments. Your revised manuscript is also likely to be sent for further evaluation by the reviewers.

**IMPORTANT - SUBMITTING YOUR REVISION**

*Re-submission Checklist*

*Published Peer Review*

*PLOS Data Policy*

*Blot and Gel Data Policy*

Sincerely,

Richard

Richard Hodge, PhD 

Associate Editor, PLOS Biology

rhodge@plos.org

REVIEWS:

Reviewer #1: In this study, Pall et. al performed a massive investigation of >40,000 RNA-seq datasets submitted to NCBI GEO and scrutinized >14,000 differential expression (DE) analyses that provided p value per feature (that is, gene). 

Properly-conducted statistical tests should yield typical distribution of p values. However, this study found that only ~25% of the examined DE analyses showed the theoretically expected shapes, indicating a widespread analytical issue in current DE studies (although monotonic improvement is observed with time). Probing the most popular DE tools (DESeq, limma, edgeR), the authors show that the shape of the p value distribution is strongly associated with the DE tool. In addition, the authors show that filtering out lowly expressed genes doubles the proportion of proper p-value distributions. Second, the fraction of genes that are truly not differentially expressed in a dataset, pi0, can be estimated from a properly-shaped p-value distribution. The authors find that in 37% of datasets this proportion is lower than 50% (that is, the majority of genes in such datasets are estimated to be genuine DE genes (DEGs)). This is in violation of a basic assumption of most DE methods according to which most of the genes in the analyzed set are not DEGs. 

The importance of studies that examine the reliability of scientific findings cannot be overstressed. However, while notwithstanding the effort and scale of the analysis presented in this study, I find that, in its current form, its impact and significance are limited. The study points to a widespread issue in the field of DE analysis. This is no doubt an interesting observation. However, to my view, this observation on its own is not sufficient to make it important/significant/useful. I would regard this observation as a first step of detecting a (potentially) important problem in the field. But there are three critical subsequent steps that are completely missing in the current study: (1) What are the consequences of the problem? Does it lead to serious biological misinterpretations of the data? Previous technical studies in this field did show how biases lead to specific functional misinterpretations. (2) What causes this bias/issue? (3) How to correct/alleviate this issue? 

These three subsequent steps are discussed in the Discussion of this study. As such the study indeed gives, as the authors put it, "a birds-eye view" on the DE field. The impact and utility of this are, to my view, limited. To make it a high-impact study speculative discussions should be replaced by solid analyses.

Few examples from the Discussion:

* Line 276: "Interestingly, limma and edgeR use very different distributional models for DE testing, supporting the notion that it might be data pre-processing, rather than the statistical test, that has the most impact on the success of DE analysis". 

- Please demonstrate this. Show how changing preprocessing (beyond filtering low expression) affect the issue. 

* Line 403: "This raises the possibility that nearly all experiments, which should have resulted in uniform distributions, were somehow shifted into other distributional classes".

- The authors can thoroughly check this by running different workflows to compare replicate samples in many datasets. Do they observe the same problem and same association with DE tool / processing pipeline? 

* Line 410. "In principle, a low PI0 could reflect true differential expression as a causal response to experimental treatment, or it could be an artefact of suboptimal data analysis."

- Again, what pi0 values do you get if you systematically apply DE analysis on replicate samples using the different tools and preprocessing pipelines? 

* Line 417. "which suggests that the low pi0-s in our dataset are indicative of problematic analytic workflows." 

- What step in the pipeline? How to alleviate? 

* Line 430. "it seems likely that many or most of the low pi0 experiments represent technical failures, most likely during data normalization." 

- This point should be strongly demonstrated rather than speculated on. 

* Line 436. "How much this level of technical malpractice influences the quality of the actual scientific conclusions - which is what really matters in the end - is an open question"

- Indeed. Without some exploration of this question, the impact of the study is of limited importance. 

* Line 438. "but one cannot but assume that this influence is substantial, and in need of corrective action."

- Agree. But a much deeper analysis should be done about the corrective actions. 

* I spotted one tangible recommendation - filtering out lowly expressed genes. This is not a novel recommendation, and it is made by most of the popular DE tools in the field. 

A typo: Line 391: refers to Fig S18. 

Reviewer #2: In this manuscript, the authors assessed inferential quality of differential expression analysis using high throughput sequencing data . They used the distribution of p-values and the proportion of nulls as two main criteria to assess the quality of differential analyses. The idea of this manuscript is interesting. However, I have the following concerns. I believe that the conclusions can be more convincing if the authors can address these concerns.

Major points:

1. The DE analysis tools used in different studies/datasets were not well clarified. In Figure 2, there are only five tools: limma, cuffdiff, deseq, edger, and unknown. I do not think only 4 methods (except "unknown") were used in all those studies. If "unknown" means all other methods, it should be named as "others" instead of "unknown". Besides, DESeq and DESeq2 use different models and should be considered two tools instead of one. The authors should replace "deseq" with a more specific names ("DESeq", "DESeq2" or "DESeq/DESeq2").

2. As the authors discussed, the non-parametric methods may be a better choice in large-sample-size differential analysis. It would be great if the authors can look into some studies/datasets using non-parametric DE analysis methods. Although text mining may be difficult, the authors can search for the keywords "Wilcoxon" and "rank-sum test" in the published papers.

3. The sample sizes of the datasets can also affect the performance of different DE tools. The sample-size effect on p-value distributions should also be evaluated.

4. The filtering step was not specified. Although we do not need to find an optimal filtering procedure in this manuscript, the authors should at least show what the threshold is, and how the threshold is decided. If multiple thresholds were tried, the results should be compared. If different thresholds were used for different species, which should be the case because different species have different numbers of genes, the different thresholds should also be listed.

5. The analysis of pi_0 is not informative or convincing. The estimated pi_0's are not the true proportions of nulls but just intermediate quantities, which are used to increase the power of identifying DE genes. Why not focusing on the proportion of DE genes (e.g. genes with FDR<0.05), which is a more straightforward statistic. 

5. In the results part, cuffdiff is different from other DE tools in almost all the analyses. The authors should provide some explanation for this phenomenon.

6. In Line 175, the authors wrote, "the increase in the fraction of anti-conservative histograms is accomplished by decreases mostly in the class of other". However, Figure 2A also showed apparent decreases in the class of bimodal and conservative. The authors should provide quantitative measurements to support their description or change it.

7. In Line 236, the results which support the conclusion that using another method does not change the results should be provided.

8. The authors concluded negative results from Line 237 to 241, but I do see associations with variables like library layout (Fig. S11). The statistical tests and corresponding p-values which support the conclusions should be included.

Minor points:

1. This manuscript only looked at the p-value-based DE tools. However, some DE tools that do not rely on p-values have been proposed (e.g. https://genomebiology.biomedcentral.com/articles/10.1186/s13059-021-02506-9) These methods can get rid of the problems regarding problematic p-values and are worth discussion.

2. The discussion section is very long. The authors may consider using subheadings.

3. Almost all main figures have overlaps with the system-generated figure numbers, making the figures hard to read. The authors should increase the figure margins.

4. Some sentences are too long to understand (e.g., the one that begins at Line 249). The authors should consider rewriting them.

---

## [Editor Report · Decision Letter 2]

10 Jan 2023

Dear Dr Maiväli,

Thank you for your patience while we considered your revised manuscript "A field-wide assessment of differential expression profiling by high-throughput sequencing reveals widespread bias" for publication as a Meta-Research Article at PLOS Biology. This revised version of your manuscript has been evaluated by the PLOS Biology editors and the Academic Editor.

Based on our Academic Editor's assessment of your revision, I am pleased to say that we are likely to accept this manuscript for publication, provided you satisfactorily address the following data and other policy-related requests that I have provided below:

(A) You may be aware of the PLOS Data Policy, which requires that all data be made available without restriction: http://journals.plos.org/plosbiology/s/data-availability. For more information, please also see this editorial: http://dx.doi.org/10.1371/journal.pbio.1001797

- Supplementary files (e.g., excel). Please ensure that all data files are uploaded as 'Supporting Information' and are invariably referred to (in the manuscript, figure legends, and the Description field when uploading your files) using the following format verbatim: S1 Data, S2 Data, etc. Multiple panels of a single or even several figures can be included as multiple sheets in one excel file that is saved using exactly the following convention: S1_Data.xlsx (using an underscore).

- Deposition in a publicly available repository. Please also provide the accession code or a reviewer link so that we may view your data before publication. 

Figure 2A-B, 3A-D, 4A-B, 5A-B, 6A-B, 7G-J, S1A-B, S2, S3, S4, S5, S6, S7, S8, S9, S10, S11, S12, S13, S14, S15, S16

(B) We note that the underlying data for the Figures may already have been provided in the data deposited at the Zenodo repository (doi:10.5281/zenodo.6795313). However, we were unable to access this data via Zenodo, so we would be grateful if you could check whether the DOI is correct or if the data has been made publicly available at this stage. 

(C) Please also ensure that figure legends in your manuscript include information on *WHERE THE UNDERLYING DATA CAN BE FOUND*, and ensure your supplemental data file/s has a legend.

(D) Please note that per journal policy, the species studied should be clearly stated in the abstract of your manuscript. 

We expect to receive your revised manuscript within two weeks. 

*Published Peer Review History*

*Press*

Sincerely,

Richard

Richard Hodge, PhD

Associate Editor, PLOS Biology

rhodge@plos.org

PLOS

---

## [Editor Report · Decision Letter 3]

20 Jan 2023

Dear Dr Maiväli,

On behalf of my colleagues and the Academic Editor, Marcus Munafo, I am pleased to say that we can accept your manuscript for publication, provided you address any remaining formatting and reporting issues. These will be detailed in an email you should receive within 2-3 business days from our colleagues in the journal operations team; no action is required from you until then. Please note that we will not be able to formally accept your manuscript and schedule it for publication until you have completed any requested changes.

PRESS

Kind regards, 

Richard

Richard Hodge, PhD

Associate Editor, PLOS Biology

rhodge@plos.org

PLOS
